# Comparing membrane sweep and cervical massage in preventing the need for formal labour induction in uncomplicated pregnancy at term: Secondary analysis of a randomized controlled trial

**Thennakoon Mudiyanselage Salila Sameera Bandara Madugalle**[1*○],
**Dissanayaka Mudiyanselage Chandana Sirimewan Jayasundara**[2○],
**Indu Asanka Jayawardane**[2○]

**1** Teaching Hospital Peradeniya, Kandy, Sri Lanka, **2** Department of Obstetrics and Gynecology, Faculty of Medicine, University of Colombo, Colombo, Sri Lanka

○ These authors contributed equally to this work.
* madugallesameera@gmail.com

## Abstract

### Introduction

Any intervention aimed at maximizing the spontaneous onset of labor and preventing formal induction will be beneficial to the client and welcomed by the provider, because it reduces postmaturity and formal labour induction.

### Methods

We recruited and randomized 312 uncomplicated singleton pregnancies at 38 weeks of gestation into three groups: membrane sweep (MS), cervical massage (CM), and sham sweep (control). Each intervention was administered at 39 weeks and repeated at 40 weeks of gestation if spontaneous labor, defined as a Modified Bishop's Score of ≥7, did not occur. (Sri Lanka clinical trials registry - SLCTR/2020/003).

### Results

The membrane sweep reduced the need for formal induction, whereas cervical massage did not. (MS vs C RR=1.4195, 95% CI=1.0326–1.9513, p=0.0310; MS vs C OR=1.8739, 95% CI=1.0664–3.2927, p=0.0290; Number Needed to Treat=7; CM vs C RR=1.2043, 95% CI=0.8598–1.6867, p=0.2795). "Survival without spontaneous labor," was lower after the membrane sweep than in controls overall (MS vs C – p=0.007; CM vs C – p=0.261), among primiparous (MS vs C p=0.047; CM vs C p=0.269) and multiparous (MS vs C p=0.038; CM vs C p=0.456) women. The membrane sweep and cervical massage were safe concerning feto-maternal complications and both reduced hospital-stay duration among multiparous women (MS vs C

---

**Data availability statement:** All relevant data are within the article and its Supporting Information files.

**Funding:** The author(s) received no specific funding for this work.

**Competing interests:** The authors have declared that no competing interests exist.

$p < 0.0001$, 95% CI = 0.5293–1.1791; CM vs C $p < 0.0001$, 95% CI = 0.6816–1.3552). There was no increased risk of emergency cesarean delivery, oxytocin augmentation, uterine hyperstimulation, postpartum bleeding, maternal pyrexia, or Apgar score < 7 at 5 minutes ($p > 0.05$). The membrane sweep was less acceptable compared to cervical massage, regardless of parity (MS vs C Primi $p = 0.001$, Multi $p = 0.0216$).

## Conclusion and recommendations

We recommend routine offer of membrane sweep to reduce the need for formal induction in term uncomplicated pregnancies, but clinicians should be aware of its inherent discomfort to women.

### Introduction

Induction of labor (IOL) should be considered when continuing the pregnancy is comparatively less beneficial than delivery. IOL is the most frequent obstetric intervention, reportedly applied in 20–25% of all pregnancies worldwide [1,2]. In Sri Lanka, the incidence is estimated to be around 32.8% by 41 weeks of gestation. IOL is associated with increased maternal complications and cesarean sections [3]. Elective inductions undertaken earlier than 39 weeks of gestation may lead to longer labors, prolonged latent stages, patient and practitioner impatience, increased costs, and higher neonatal morbidity [4,5]. IOL medicalizes the birthing process, impacting healthcare system resources and negatively affecting the birth experience for the family.

Any simple, harmless, and effective intervention that promotes the spontaneous onset of labor would be highly beneficial. It would help prevent complications associated with formal induction and allow the mother to experience a more natural childbirth with minimal interventions. Therefore, the National Institute for Health and Care Excellence (NICE) recommends offering a membrane sweep to all suitable pregnant women at 39 weeks, with additional sweeps as needed if labor does not commence. It also recommends cervical massage if the examining finger cannot pass through the closed cervical os. [6].

This recommendation is based on the premise that cervical massage is as effective as a membrane sweep. However, there is no well-conducted research designed to compare the intervention-specific effects and its utility either as a method of induction or as an adjunct to prevent formal induction. Moreover, available physiological studies raise serious concerns.

Both the membrane sweep and cervical massage are based on the premise that they induce a local release of prostaglandins. This has been evidenced by in-vitro stimulation studies on strips of cervical tissue [7,8]. Prostaglandins interact complexly with cervical cells and the extracellular matrix. Their local concentrations lead to cervical softening, shortening, and dilation. Additionally, prostaglandins stimulate myometrial receptors, triggering rhythmic uterine contractions.

Fetal membranes and the decidua are critical sites for intrauterine prostaglandin synthesis. These mechanical methods cause disruption at these sites, leading to

prostaglandin synthesis and release [9,10]. The degree of disruption is important, as shown by the work of Marc J.N.C et al., who compared extra-amniotic Foley catheter induction with and without the instillation of cellulose gel. The cellulose gel, which had a much greater extra-amniotic spread, separated a correspondingly larger area of chorio-amnion from the decidual surface compared to just the inflated balloon of a Foley catheter. This resulted in a greater rise in prostaglandin levels in peripheral blood at 5 minutes, which gradually reduced over the next 12 hours [11]. This finding correlated with the work of Manabe et al., which suggested that a larger volume balloon may produce greater separation and a larger rise in prostaglandin levels. Additionally, a larger balloon would stretch the cervix more [12].

A well-conducted membrane sweep, which separates a significantly larger area of the chorio-amnion from the decidua and stretches the cervix more, likely causes a much greater rise in prostaglandin levels compared to a cervical massage. The cervical massage is limited to external stimulation of the cervix without any disturbance of the chorio-decidual interface. Additionally, the membrane sweep stretches the cervix, triggering the Ferguson reflex, which leads to pulsatile oxytocin secretion from the posterior pituitary, unlike the cervical massage [13].

The National Institute for Health and Care Excellence (NICE) now recognizes the balloon catheter as a mechanical induction method. There are variants, such as the double balloon and single balloon, but evidence is limited in determining the superiority of one over the other [14]. Currently, the membrane sweep is considered an adjunct to the induction process rather than a standalone mechanical induction method.

Although the membrane sweep is widely applied in Sri Lanka, cervical massage has not gained much popularity, and neither method has been standardized or well described in the literature. Therefore, we conducted a randomized controlled trial to compare the intervention-specific efficacy of the membrane sweep and cervical massage in preventing the formal induction of labor.

## Methods

A randomized, single-blind controlled trial was conducted at De Soysa Hospital for Women, Colombo 08, Sri Lanka, from 1st of February to 31st of July, 2020 (Trial Registration No: SLCTR/2020/003, Date of registration: 22nd January, 2020). A total of 349 uncomplicated singleton pregnancies between 38–40 weeks of gestation were selected via simple random sampling from the antenatal clinic at the Professorial Obstetrics and Gynaecology Unit. Sample size estimation was based on prior research indicating an increase in spontaneous labor from 75% to 90% following membrane sweeping (Alpha = 0.05, Beta = 0.8, Minimum sample size per group = 100). Thirty-seven women were excluded due to predefined eligibility criteria (Tables 1 and 2).

312 prospective participants were provided, written information with verbal clarifications where necessary with one week period to consider participation. The principal investigator was available for consultation over the phone during that one week, during work hours and weekend. At the end of the 7-day period they were re-interviewed, further clarifications were provided as required, and consent forms were signed, in the presence of the principal investigator. All 312 women were recruited into the study and randomized into three groups using a random number sequence generated using Research Randomizer (https://www.randomizer.org). Sequentially numbered, equal-sized red color cards with the

**Table 1.  Inclusion criteria.**

| Inclusion criteria |
| --- |
| 1. 38–40 completed weeks of gestation, |
| 2. No clear indication for early delivery, emergency, or elective cesarean delivery |
| 3. Had normal fetal heart rate rhythm on CTG, normal fetal dopplers on ultrasound scan, adequate growth for gestational age, |
| 4. Modified Bishops' Score ≤6, but the examining finger was admissible into the cervix during pelvic examination, so all interventions were equally applicable, |

**Table 2. Exclusion criteria.**

| Exclusion criteria |
| --- |
| 1. Intra uterine fetal demise, |
| 2. Ruptured membranes |
| 3. Abnormal amniotic fluid index, |
| 4. Fetal anomalies |
| 5. Abnormal fetal growth – growth restriction or macrosomia |
| 6. Any contraindication for induction of labor |
| 7. Latex allergies (Latex gloves were worn during certain procedures during the study). |
| 8. Tightly closed cervix where membrane sweep would not be possible |

**Table 3. Interventions and pre-intervention monitoring.**

| Interventions | | Peri-intervention monitoring |
| --- | --- | --- |
| Membrane sweep | Insertion of one examining finger through the internal cervical OS and performing three 360° sweeps separating the chorio-amnion from the lower uterine segment over 15–30 seconds | • Admitted for 24 hours for intervention and monitoring.<br>• linical assessment of the woman to confirm maternal well-being.<br>• TG and ultrasound scans to confirm fetal well-being before intervention.<br>• CTG 2 hours after intervention |
| Cervical massage | Three 360° sweeps of two examining fingers during vaginal examination massaging the outside of the cervix along the fornixes for 15–30 seconds. | • Maternal pulse rate, blood pressure, temperature, fetal heart rate by auscultation every 4 hours till 24 hours after intervention. |

allocated intervention printed on the reverse side, were stored in an opaque container until the participants signed the consent forms. Randomization was performed by the three investigators. The first group received a membrane sweep (MS), the second group received a Cervical Massage (CM), and the third group, serving as the control, underwent a sham sweep following a pelvic examination to record the Bishop's score at recruitment.

Interventions were administered at 39 weeks of gestation, and repeated at 40 weeks if spontaneous labor did not occur within one week (Table 3). Stringent safety protocols were implemented to withdraw participants in cases of latex allergies, cervical tears, vasovagal attacks, uterine rupture, maternal or fetal infections, or any psychological concerns. All adverse events were managed per standard unit protocols, serious adverse events were reported to a Data and Safety Monitoring Board composed of two external obstetricians and an external biostatistician.

Participants were blinded to the specific intervention received, and the intervention acceptability was assessed. All interventions, being vaginal, were performed during digital vaginal examination with prior consent, and no additional blinding methods were utilized. The principal investigator provided counselling based on a template and performed all interventions to ensure uniformity.

Participants were followed up every third day as outpatients until 40 + 6 weeks of gestation. A Modified Bishop's Score of 6 or less indicated no spontaneous labor, prompting formal induction, using either a Foley catheter or prostaglandin vaginal tablet, as an inpatient. Participants going into spontaneously labor, i.e., achieving a Modified Bishop's Score of 7 or more, were admitted for labour management. All participants were managed according to standard unit protocols and monitored according to the study protocol till 24 hours postpartum for complications (Table 4).

Maternal acceptability was assessed using a pre-validated, interviewer-administered questionnaire, consisting of four Likert scale items. This questionnaire, developed by the principal author, is an adaptation of the seven-fold conceptual framework for the acceptability of a healthcare intervention published by Sekhon et al. The seven components are

**Table 4. Outcome measures.**

| |
|---|
| **Overall outcome measures** |
| • Percentage of women going into spontaneous labour after membrane sweep and cervical massage. |
| • Survival" (proportion of subjects without spontaneous labour) after membrane sweep and cervical massage. |
| • Maternal acceptability of membrane sweep and cervical massage |
| **Specific outcome measures – labour outcome measures** |
| • Emergency Cesarean delivery rates after spontaneous labour following membrane sweep and cervical massage. |
| • Percentage of Oxytocin augmentation during labour following membrane sweep and cervical massage. |
| • Uterine hyperstimulation rates following membrane sweep and cervical massage. |
| **Specific outcome measures – Maternal outcome measures** |
| • Postpartum bleeding rates following membrane sweep and cervical massage. |
| • Maternal pyrexia rates following membrane sweep and cervical massage. |
| • Duration of hospital stay following membrane sweep and cervical massage. |
| **Specific outcome measures – Neonatal outcome measures** |
| • APGAR at 5 minutes following membrane sweep and cervical massage. |
| • Rate of APGAR <7 at 5 minutes following membrane sweep and cervical massage |

affective attitude, burden, perceived effectiveness, self-efficacy, ethicality, intervention coherence, and opportunity cost. The first five components were extracted due to their patient-focused nature [15].

The four-item questionnaire was developed using five-step pre-validation method for patient-reported outcome instruments, published by Prior et al., [16] and the discriminant content validity method for theory-based instruments, published by Johnston et al., [17]. It was assessed for answerability, comprehensibility, and relevance using adaptations from the Think-Aloud studies published by French et al., [18], and Green and Gilhooly [19]. The resultant questionnaire was validated in a pilot study with 202 pregnant women selected using the same exclusion and inclusion criteria as the current study (S-CVI = 0.875, Cronbach's alpha = 0.876).

Given the temporal evolution of acceptability from before, during, and after the intervention described by Sekhon et al, the questionnaire was administered 24 hours postnatally to capture an overall impression. Each item was scored from 1 to 5, yielding a total score for each participant.

Data was analyzed using SPSS 26. Comparisons between the control and the intervention groups were performed separately. Demographic characteristics, mean acceptability scores, duration of hospital stay, and mean Apgar score at 5 minutes of the three groups were compared using t-tests to and P values with 95% confidence intervals were reported.

Spontaneous labor rates, emergency cesarean delivery rates, postpartum hemorrhage rates, maternal pyrexia rates, and rates of neonates with an Apgar score of less than 7 at 5 minutes after birth were compared using proportions and the Chi-square test, and P values with 95% confidence intervals were reported. Kaplan-Meier survival plots for proportion of subjects without spontaneous labour, after membrane sweep, cervical massage, and controls were generated. The Log-Rank test compared survival distributions between intervention groups and controls separately. P values of <0.05 were deemed statistically significant.

Formal induction rates after membrane sweep and cervical massage were compared to controls individually, with odds ratios (OR), relative risk ratios (RR), Control and experiment event rates, attributable risk ratio, and number needed to treat (NNT) for each intervention calculated using contingency tables. Subgroup analysis was conducted based to parity.

The study was reviewed and approved by the Ethics Review Committee of the Faculty of Medicine, University of Colombo on October 17, 2019, for application no. PROTOCOL-019-064.

 

## Results

Out of the 349 women selected, 1 had a fetus with a cardiac anomaly, 3 had abnormal liquor, and 7 had small-for-gestational-age or growth-restricted fetuses. During the initial pelvic examination, it was not possible to insert the examining finger into the cervical canal due to tightly closed cervices in 26 women, making membrane sweep impossible in these women. The Cervical Massage (CM), Membrane Sweep (MS), and Control (C) groups respectively had 106, 104, and 102 subjects. One participant in the CM group, 2 in the MS group, and 1 in the C group withdrew consent. Two women in the MS group and 3 in the CM group delivered at other hospitals and were excluded from the study. Therefore, 100, 102, and 101 subjects in the MS, CM, and C groups respectively were included in the final analysis according to the originally assigned groups (Fig 1). All three groups had similar demographic characteristics (Table 5). All interventions were promptly administered without any decision-to-intervention delay. All participants had satisfactory maternal and fetal conditions before the interventions.

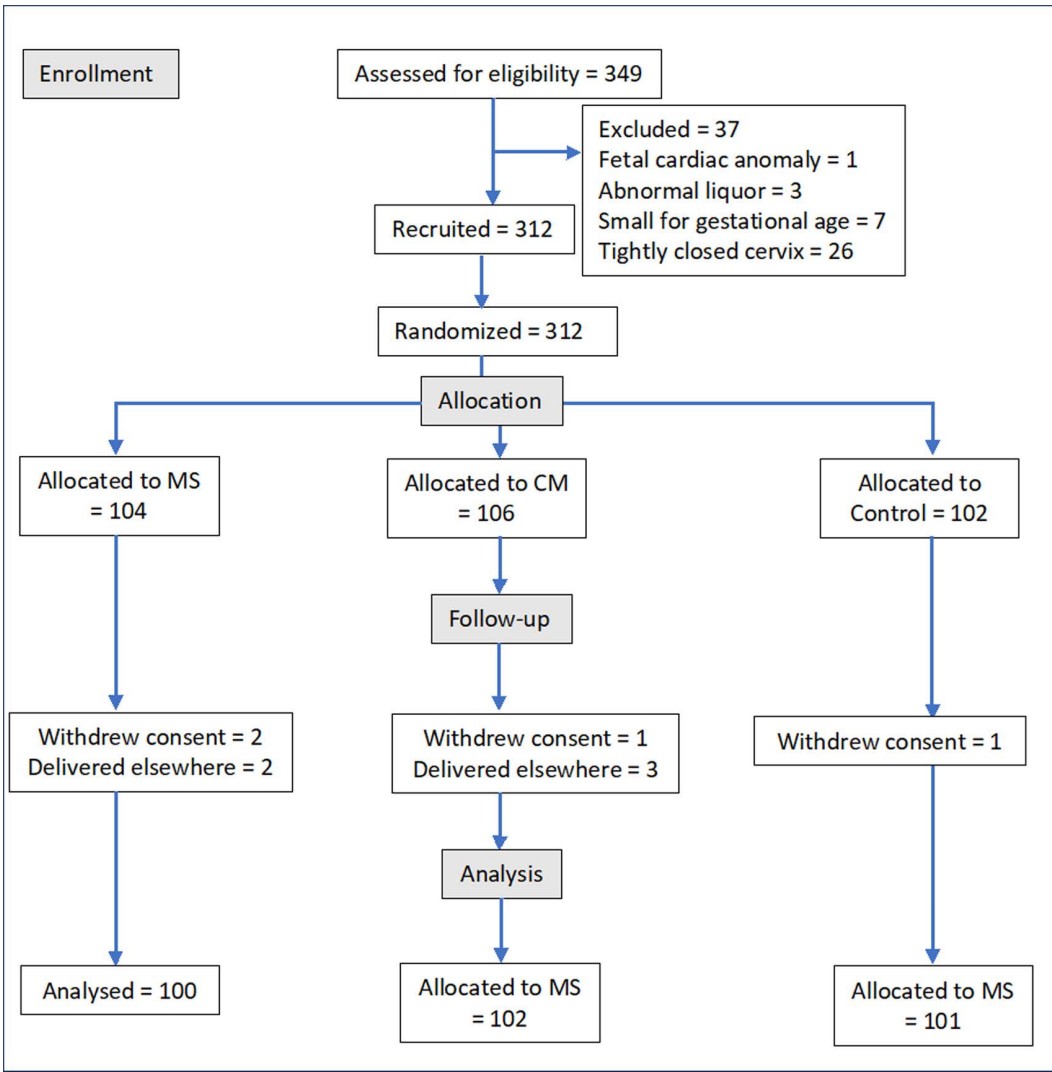

**Fig 1.  CONSORT flowchart.** (Membrane Sweep – MS, Cervical Massage – CM, Control -C).

**Table 5. Demographic characteristics (Membrane Sweep – MS, Cervical Massage – CM, Control -C).**

| Demographic Characteristic | MS group N = 100 | | CM group N = 102 | | C group N = 101 | |
|---|---|---|---|---|---|---|
| | Primi | Multi | Primi | Multi | Primi | Multi |
| Number | 53 | 47 | 61 | 41 | 54 | 47 |
| Age | | | | | | |
| Mean | 26.91 | 30.26 | 26.75 | 29.95 | 26.82 | 30.68 |
| SD | 2.574 | 3.200 | 2.737 | 3.827 | 2.860 | 3.401 |
| *P* compared to C group | 0.8646 | 0.5390 | 0.8936 | 0.3461 | | |
| Modified Bishop's score at recruitment | | | | | | |
| Mean | 2.87 | 4.55 | 2.85 | 4.51 | 2.72 | 4.70 |
| SD | 1.057 | 0.880 | 1.078 | 0.746 | 0.998 | 0.750 |
| *P* compared to C group | 0.4520 | 0.3158 | 0.5054 | 0.2376 | | |

Spontaneous labor (i.e., prevention of formal induction) occurred in 52% of cases after MS, 44.12% after CM, and 36.63% after the sham sweep. The comparisons showed:

- MS vs C: Relative Risk (RR) = 1.4195, 95% CI = 1.0326–1.9513, p = 0.0310

- MS vs C: Odds Ratio (OR) = 1.8739, 95% CI = 1.0664–3.2927, p = 0.0290

- MS vs C: Number needed to treat (NNT) = 7

- CM vs C: RR = 1.2043, 95% CI = 0.8598–1.6867, p = 0.2795

Subgroup analysis based on parity did not show any statistically significant difference between any methods..

The duration from intervention to diagnosis of spontaneous labor was assessed. Kaplan-Meier survival analysis showed that after a membrane sweep, the proportion of subjects without spontaneous labor was significantly lower compared to controls, regardless of parity. Membrane sweep caused women to undergo spontaneous labor earlier than the controls, while cervical massage did not (Tables 6 and 7).

Analysis of survival plots revealed another interesting trend. Primiparous women required two interventions one week apart for a significant effect, while a single administration sufficed for multiparous women (Figs 2 and 3).

The maternal acceptability score of cervical massage was significantly higher than that of the membrane sweep, regardless of parity. Particularly in primiparous women, the difference in acceptability between cervical massage and membrane sweep was highly significant (p = 0.0001) (Table 8).

Emergency Cesarean delivery rates, oxytocin augmentation rates, and uterine hyperstimulation rates were similar in the MS, CM, and C groups during the overall analysis as well as the subgroup analysis according to parity (Table 9).

There was no difference in postpartum bleeding and maternal pyrexia rates between the MS, CM, and C groups (Table 10). In contrast, the duration of hospital stay was significantly shorter in women who had either a membrane sweep (MS) or cervical massage (CM) overall, compared to controls (MS vs C – p = 0.0015, 95% CI = 0.1752–0.7166, CM vs C – p = 0.0197, 95% CI = 0.0603–0.6757).

Both membrane sweep and cervical massage shortened hospital stay in multiparous women only (MS vs C – p < 0.0001, 95% CI = 0.5293–1.1791, CM vs C – p < 0.0001, 95% CI = 0.6816–1.3552). Interestingly, there was no such difference among primipara. APGAR value at 5 minutes was assessed, and all three methods had similar values. There were no neonates with an APGAR <7 at 5 minutes among study participants in any group (p > 0.05, Table 10).

The trial concluded with all participants progressing through the study protocol without any significant adverse outcomes. The study protocol and the anonymized data sets are freely accessible under the CC By 4.0 license: S1 Data– questionnaire data, S2 Data set Multi, S3 Data set Primi, S4 Data– Study protocol [20–22].

**Table 6. Contingency table for spontaneous labour for Membrane Sweep (MS), Cervical Massage (CM), and Control (C) groups.**

| Intervention group | Number of women who underwent Spontaneous labour | | | Number of women who Needed formal induction | | |
|---|---|---|---|---|---|---|
| | Overall | Primi | Multi | Overall | Primi | Multi |
| MS | 52 (52%) | 28 (52.83%) | 24 (51.06%) | 48 (48%) | 25 (47.17%) | 23 (48.93%)_ |
| CM | 45 (44.12%) | 28 (45.9%) | 17 (41.46%) | 57 (55.88%) | 33 (54.09%) | 24 (58.53%) |
| C | 37(36.63%) | 21 (38.89%) | 16 (34.04%) | 64 (63.37%) | 33 (61.11%) | 31 (65.95%) |

**Table 7. P-values from Log rank tests for comparison of survival curves after membrane sweep versus cervical massage (MS), Cervical massage (CM) and Controls (C).**

| Parity | Log Rank probability values comparing survival plots for interventions | |
|---|---|---|
| | MS vs C | CM vs C |
| Primiparous | *P*=0.047 | *P*=0.269 |
| Multiparous | *p*=0.038 | *P*=0.456 |

## Discussion

Induction of labor is currently an in-patient procedure under existing protocols, with only a few units practicing outpatient formal labor induction. Reducing the need for formal induction is highly attractive to pregnant women, obstetricians, and policymakers, as it would lead to significant cost savings, minimal disruptions to the daily lives of pregnant women and their families, and an improved birth experience.

Membrane sweep (MS) was more successful in preventing formal induction compared to controls, whereas no such difference was noted after CM. "Survival analysis" clearly shows that MS was more effective due to the lower numbers of women without spontaneous labor. Since we selected a population where both membrane sweep and cervical massage were possible, we were able to determine the intervention-specific effect on the cervix, allowing for an evidence-based decision when selecting an induction prevention method.

Our findings correlate with the available physiological understanding of cervical ripening and the process of induction of labor. The magnitude of local prostaglandin release depends on the degree of disruption of the chorio-decidual interface and the degree of cervical stretching [23,24]. Membrane sweep disrupts the chorio-decidual interface to a much higher degree compared to cervical massage, where there is no such disruption. The insertion of the examining finger stretches the cervix, increasing local prostaglandin release as well as triggering the Ferguson reflex.

Available literature does not suggest any increased risk of infection during either membrane sweep or cervical massage. Admittedly, the evidence base for cervical massage is quite poor [25].

In an interesting turnout, we found that primiparous women needed two sweeps one week apart to prevent formal induction compared to the single sweep in multiparous women. This may be due to the higher receptiveness and sensitivity of the multiparous cervix and/or the higher initial Modified Bishop's Score. Since this study was not designed to compare different regimens of membrane sweep, we cannot recommend one regimen over the other. There are a few studies looking into various sweep regimens. Putnam et al. compared a single sweep to weekly sweeps and found no significant difference in rates of spontaneous labor [26]. Another study compared daily sweeps to vaginal Dinoprostone and found that sweeps were more effective and less costly [27]. The authors are conducting ongoing research to compare different regimens.

For multiparous women, both membrane sweep and cervical massage significantly reduced the duration of hospital stay compared to controls in our study. Our data supports the safety profile of both membrane sweep and cervical massage without any added risks. Neonatal parameters also show excellent tolerability of these mechanical methods, though there is no long-term data on the well-being of babies born following the use of membrane sweep or cervical massage.

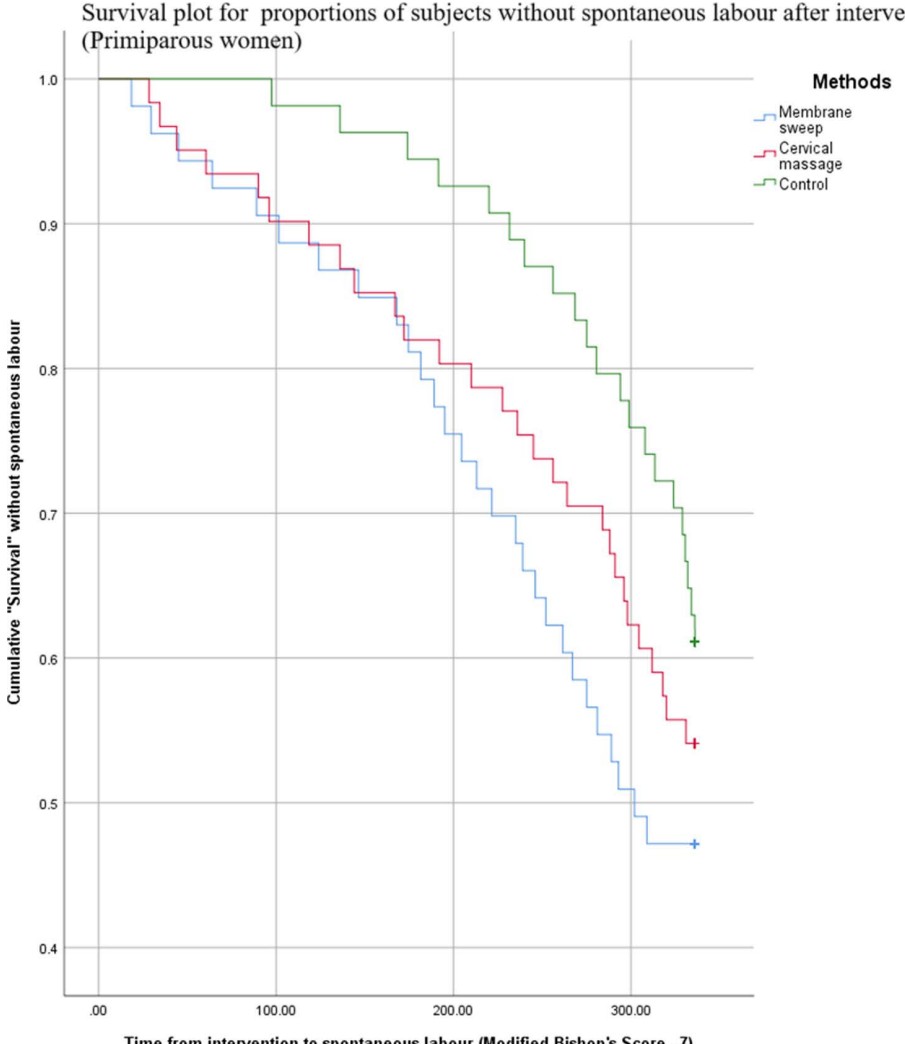

**Fig 2. Survival plot for proportion of subjects without spontaneous labour - primiparous women.** This figure demonstrates changes of the cumulative percentage of primiparous women who remain without going into spontaneous labour plotted against the time from initial intervention.

Both membrane sweep and cervical massage were very safe procedures with minimal adverse events according to our study. The routine practice worldwide is to perform these as outpatient procedures, and our findings reinforce this practice. Admission and monitoring were conducted for research purposes and to objectively assess the safety of these interventions.

To ensure the largest possible prostaglandin release, we recommend the technique for membrane sweeping we adopted for the study. The examining finger should be inserted through the internal cervical os, followed by three circumferential sweeps between the fetal membranes and the decidua, separating the two. The depth of separation should be the length of the examining finger. While one sweep separates the chorio-amnion from the decidua, the two additional sweeps exert a stretching effect on the cervix, maximizing prostaglandin release and triggering the Ferguson reflex.

In conclusion, we recommend the routine administration of a membrane sweep to prevent formal induction in term uncomplicated pregnancies. However, clinicians should be aware of its inherent discomfort and should counsel women about this during delivery planning.

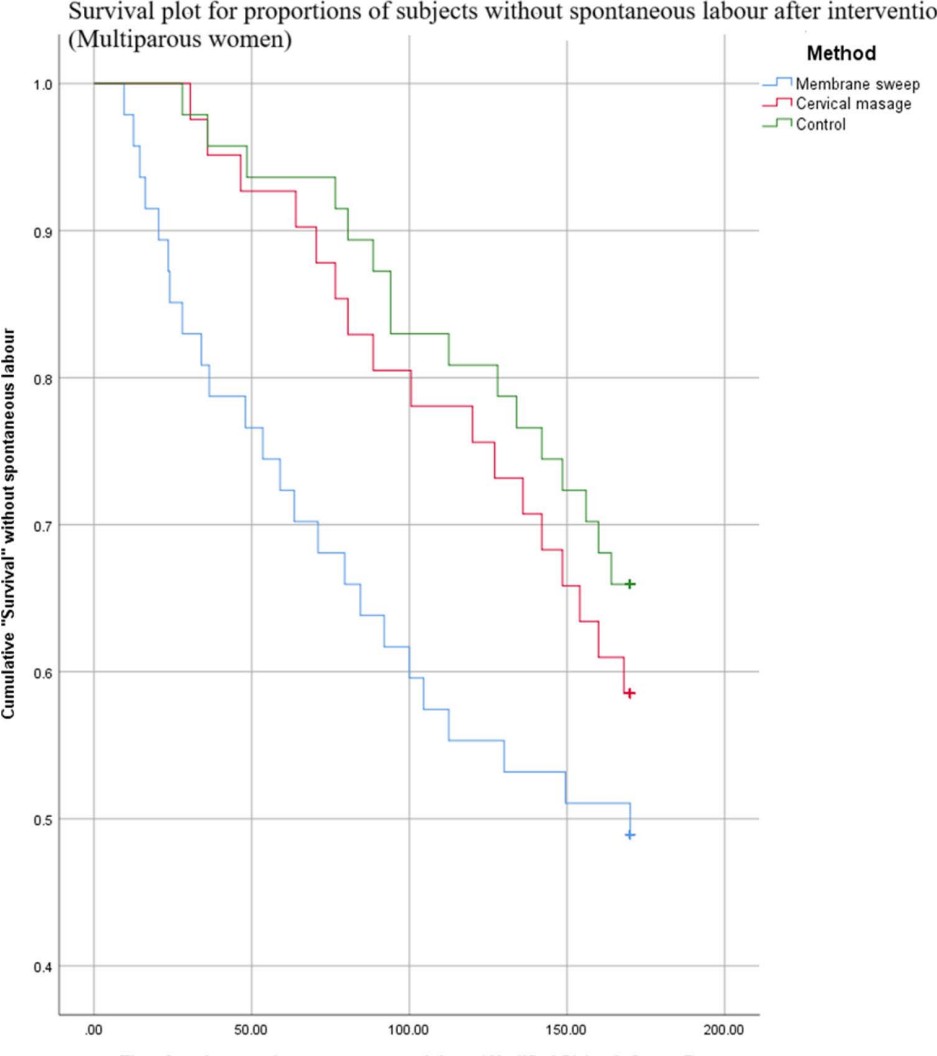

**Fig 3. Survival plot for proportion of subjects without spontaneous labour - Multiparous women.** This figure demonstrates changes of the cumulative percentage of multiparous women who remain without going into spontaneous labour plotted against the time from initial intervention.

**Table 8. Maternal acceptability scores for membrane sweep (MS) and cervical massage (CM) groups.**

| Method | Membrane sweep (n – 100) | | Cervical Massage (n- 102) | | P value |
|---|---|---|---|---|---|
| | Mean score | SD | Mean score | SD | |
| Parity | | | | | |
| Primiparous  (n - 53) | 8.6415 | 2.4702 | 11.5082 | 3.7755 | 0.0001 |
| Multiparous  (n – 47) | 14.5957 | 2.7556 | 15.8780 | 2.2359 | 0.0216 |

**Table 9. Labour outcome rates between membrane sweep (MS), Cervical Message (CM), and Control (C) groups.**

| Labour Outcome | MS | | | CM | | | C | | |
|---|---|---|---|---|---|---|---|---|---|
| | Overall N=52 | Primi N=28 | Multi N=24 | Overall N=45 | Primi N=28 | Multi N=17 | Overall N=37 | Primi N=21 | Multi N=16 |
| Oxytocin augmentation | | | | | | | | | |
| Rate | 42/52 | 25/28 | 17/24 | 41/45 | 27/28 | 14/17 | 35/37 | 21/21 | 14/16 |
| Percentage | 80.8% | 89.3% | 70.8% | 91.1% | 96.4% | 82.3% | 94.6% | 100% | 87.5% |
| P value (vs C) | 0.0614 | 0.1255 | 0.2220 | 0.5496 | 0.3866 | 0.6847 | | | |
| Uterine hyperstimulation | | | | | | | | | |
| Rate | 6/52 | 3/28 | 3/24 | 3/45 | 1/28 | 2/17 | 3/37 | 2/21 | 1/16 |
| Percentage | 11.5% | 10.7% | 12.5% | 6.7% | 3.6% | 11.8% | 8.1% | 9.5% | 6.2% |
| P value (vs C) | 0.5989 | 0.8928 | 0.5239 | 0.8044 | 0.3947 | 0.5879 | | | |
| Emergency Cesarean rate | | | | | | | | | |
| Rate | 5/52 | 4/28 | 1/24 | 6/45 | 5/28 | 1/17 | 1/37 | 1/21 | 0/16 |
| Percentage | 9.6% | 14.3% | 4.2% | 13.3% | 17.8% | 5.9% | 2.7% | 4.8% | 0% |
| P Value (vs C) | 0.2020 | 0.2804 | 0.4140 | 0.0884 | 0.1706 | 0.3321 | | | |

**Table 10. Maternal and neonatal outcomes in Membrane sweep (MS), Cervical massage (CM), and Control (C) groups.**

| Maternal and Neonatal Outcome | MS | | | CM | | | C | | |
|---|---|---|---|---|---|---|---|---|---|
| | Overall N=52 | Primi N=28 | Multi N=24 | Overall N=45 | Primi N=28 | Multi N=17 | Overall N=37 | Primi N=21 | Multi N=16 |
| Duration of hospital stay | | | | | | | | | |
| Mean | 2.5 | 2.536 | 2.428 | 2.578 | 2.75 | 2.294 | 2.946 | 2.667 | 3.312 |
| SD | 0.610 | 0.693 | 0.509 | 0.723 | 0.799 | 0.470 | 0.664 | 0.658 | 0.479 |
| P (vs C) | 0.0015 | 0.5068 | <0.0001 | 0.0197 | 0.6993 | <0.0001 | | | |
| Postpartum bleeding | | | | | | | | | |
| Rate Percentage | 0/52 0% | 0/28 0% | 0/24 0% | 2/45 4.4% | 1/28 3.6% | 1/17 5.9% | 0/37 0% | 0/21 0% | 0/16 0% |
| P (vs C) | 1 | 1 | 1 | 0.1972 | 06492 | 03321 | | | |
| Maternal pyrexia | | | | | | | | | |
| Rate Percentage | 4/52 7.7% | 2/28 7.1% | 2/24 8.3% | 0/45 0% | 0/28 0% | 0/17 0% | 1/37 2.7% | 1/21 4.8% | 0/16 0% |
| P (vs C) | 0.3136 | 0.7335 | 0.2422 | 0.2703 | 0.2463 | 1 | | | |
| APGAR at 5 minutes | | | | | | | | | |
| Mean SD | 9.827 0.550 | 9.893 0.416 | 9.75 0.676 | 9.822 0.576 | 9.786 0.630 | 9.882 0.485 | 9.865 0.585 | 9.905 0.436 | 9.812 0.75 |
| P (vs C) | 0.5772 | 0.9231 | 0.7612 | 0.5737 | 0.4616 | 0.6946 | | | |
| APGAR <7 at 5 minutes | | | | | | | | | |
| Rate | 0/52 | 0/28 | 0/24 | 0/45 | 0/28 | 0/17 | 0/37 | 0/21 | 0/16 |
| Percentage | 0% | 0% | 0% | 0% | 0% | 0% | 0% | 0% | 0% |
| P (vs C) | 1 | 1 | 1 | 1 | 1 | 1 | | | |

## Supporting information

**S1 Data. Questionnaire data.**
(XLSX)

**S2 Data. Data set multi.**
(XLSX)

**S3 Data. Data set primi.**
(XLSX)

**S4 Data. Protocol Version 2.1.**
(XLSX)

**S5 Data. Consort 2010 checklist.**
(DOCX)

## Acknowledgments

Special thanks to Dr. Muthukuda C. for invaluable advice on statistical analysis.

## Author contributions

**Conceptualization:** Thennakoon Mudiyanselage Salila Sameera Bandara Madugalle, Dissanayaka Mudiyanselage Chandana Sirimewan Jayasundara.

**Data curation:** Thennakoon Mudiyanselage Salila Sameera Bandara Madugalle, Dissanayaka Mudiyanselage Chandana Sirimewan Jayasundara, Indu Asanka Jayawardane.

**Formal analysis:** Thennakoon Mudiyanselage Salila Sameera Bandara Madugalle, Dissanayaka Mudiyanselage Chandana Sirimewan Jayasundara, Indu Asanka Jayawardane.

**Investigation:** Thennakoon Mudiyanselage Salila Sameera Bandara Madugalle, Dissanayaka Mudiyanselage Chandana Sirimewan Jayasundara, Indu Asanka Jayawardane.

**Methodology:** Thennakoon Mudiyanselage Salila Sameera Bandara Madugalle, Dissanayaka Mudiyanselage Chandana Sirimewan Jayasundara, Indu Asanka Jayawardane.

**Project administration:** Thennakoon Mudiyanselage Salila Sameera Bandara Madugalle, Dissanayaka Mudiyanselage Chandana Sirimewan Jayasundara, Indu Asanka Jayawardane.

**Resources:** Thennakoon Mudiyanselage Salila Sameera Bandara Madugalle.

**Software:** Thennakoon Mudiyanselage Salila Sameera Bandara Madugalle.

**Supervision:** Dissanayaka Mudiyanselage Chandana Sirimewan Jayasundara, Indu Asanka Jayawardane.

**Visualization:** Thennakoon Mudiyanselage Salila Sameera Bandara Madugalle.

**Writing – original draft:** Thennakoon Mudiyanselage Salila Sameera Bandara Madugalle, Dissanayaka Mudiyanselage Chandana Sirimewan Jayasundara.

**Writing – review & editing:** Thennakoon Mudiyanselage Salila Sameera Bandara Madugalle, Dissanayaka Mudiyanselage Chandana Sirimewan Jayasundara, Indu Asanka Jayawardane.

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
