## [Decision Letter · Decision Letter 0]

10 Jan 2025

PONE-D-24-26866Comparing membrane sweep and cervical massage in preventing the need for formal labour induction in uncomplicated pregnancy at term: Secondary analysis of a randomized controlled trial.PLOS ONE

Dear Dr. Madugalle,

Thank you for submitting your manuscript to PLOS ONE. After careful consideration, we feel that it has merit but does not fully meet PLOS ONE’s publication criteria as it currently stands. Therefore, we invite you to submit a revised version of the manuscript that addresses the points raised during the review process.

We look forward to receiving your revised manuscript.

Kind regards,

Nishel Mohan Shah, PhD

Academic Editor

PLOS ONE

Journal Requirements:

2. We note that the original protocol file you uploaded contains a confidentiality notice indicating that the protocol may not be shared publicly or be published. Please note, however, that the PLOS Editorial Policy requires that the original protocol be published alongside your manuscript in the event of acceptance. Please note that should your paper be accepted, all content including the protocol will be published under the Creative Commons Attribution (CC BY) 4.0 license, which means that it will be freely available online, and any third party is permitted to access, download, copy, distribute, and use these materials in any way, even commercially, with proper attribution.

Therefore, we ask that you please seek permission from the study sponsor or body imposing the restriction on sharing this document to publish this protocol under CC BY 4.0 if your work is accepted. We kindly ask that you upload a formal statement signed by an institutional representative clarifying whether you will be able to comply with this policy. Additionally, please upload a clean copy of the protocol with the confidentiality notice (and any copyrighted institutional logos or signatures) removed.

3. We are unable to open your Supporting Information file [Data set primi.sav and Data set multi.sav]. Please kindly revise as necessary and re-upload.

Please confirm at this time whether or not your submission contains all raw data required to replicate the results of your study. Authors must share the “minimal data set” for their submission. PLOS defines the minimal data set to consist of the data required to replicate all study findings reported in the article, as well as related metadata and methods (https://journals.plos.org/plosone/s/data-availability#loc-minimal-data-set-definition ).

If your submission does not contain these data, please either upload them as Supporting Information files or deposit them to a stable, public repository and provide us with the relevant URLs, DOIs, or accession numbers. For a list of recommended repositories, please see https://journals.plos.org/plosone/s/recommended-repositories .

5. Please amend your authorship list in your manuscript file to include authors Dr. Thennakoon Mudiyanselage Salila Sameera Bandara Madugalle.

7. "Please include captions for your Supporting Information files at the end of your manuscript, and update any in-text citations to match accordingly. Please see our Supporting Information guidelines for more information: http://journals.plos.org/plosone/s/supporting-information .

Additional Editor Comments:

Dear authors,

Thank you for submitting this paper. The reviewers have recommended minor revisions and have outlined a series of comments to improve your work. Please go through these comments and we will be happy to review a revised manuscript.

Yours sincerely,

Nishel

Reviewer 1:

This manuscript reports secondary analysis results of a randomized controlled clinical trial investigating membrane sweep and cervical massage in preventing the need for formal

labour induction in uncomplicated pregnancy at term. My comments are listed in the following.

In the Method section, please provide descriptions of statistical methods used for comparing outcome rates between groups.

Please correct “long-Rank mantel-Cox test” as “log-rank test”. Please note that it’s log-rank instead of long-rank. Since “log-rank test” is sometimes called “Mantel-Cox test”, you don’t have to put both names together.

Line 220-221, What’ RR and OR? How were they calculated? Please define acronyms the first time they appear in the text.

In the results, it is better to use “RR=” and “95% CI=” rather than “RR-” or “95% CI-”.

Line 255, “due to small subgroup size” is not the reason for insignificant difference since multiparous group has even smaller size but with significant difference.

In table 10, please add p values for each outcome. Those “0” p-values for comparing 0% vs. 0% are not correct. They should be close to 1. It’s better to ask a biostatistician to guide and help with the data analysis.

Reviewer 2:

Abstract is longer than recommended 300 words and would benefit from having headings (introduction, methodology, results, conclusion).

Some grammar amendments to improve readability such as:

105- '...a far larger area of chorio-amnion off the decidua...'. The word 'off' should be replaced by 'of'. This is a recurrent theme throughout the manuscript.

135- After first coma should be 'and' or if coma is kept 'they'.

Reviewers' comments:

Reviewer's Responses to Questions

**Comments to the Author**

1. Is the manuscript technically sound, and do the data support the conclusions?

Reviewer #1: Partly

Reviewer #2: Yes

2. Has the statistical analysis been performed appropriately and rigorously? 

Reviewer #1: N/A

Reviewer #2: Yes

3. Have the authors made all data underlying the findings in their manuscript fully available?

Reviewer #1: No

Reviewer #2: Yes

4. Is the manuscript presented in an intelligible fashion and written in standard English?

Reviewer #1: Yes

Reviewer #2: No

5. Review Comments to the Author

Reviewer #1: This manuscript reports secondary analysis results of a randomized controlled clinical trial investigating membrane sweep and cervical massage in preventing the need for formal

labour induction in uncomplicated pregnancy at term. My comments are listed in the following.

In the Method section, please provide descriptions of statistical methods used for comparing outcome rates between groups.

Please correct “long-Rank mantel-Cox test” as “log-rank test”. Please note that it’s log-rank instead of long-rank. Since “log-rank test” is sometimes called “Mantel-Cox test”, you don’t have to put both names together.

Line 220-221, What’ RR and OR? How were they calculated? Please define acronyms the first time they appear in the text.

In the results, it is better to use “RR=” and “95% CI=” rather than “RR-” or “95% CI-”.

Line 255, “due to small subgroup size” is not the reason for insignificant difference since multiparous group has even smaller size but with significant difference.

In table 10, please add p values for each outcome. Those “0” p-values for comparing 0% vs. 0% are not correct. They should be close to 1. It’s better to ask a biostatistician to guide and help with the data analysis.

Reviewer #2: Abstract is longer than recommended 300 words and would benefit from having headings (introduction, methodology, results, conclusion).

Some grammar amendments to improve readability such as:

105- '...a far larger area of chorio-amnion off the decidua...'. The word 'off' should be replaced by 'of'. This is a recurrent theme throughout the manuscript.

135- After first coma should be 'and' or if coma is kept 'they'.

6. PLOS authors have the option to publish the peer review history of their article (what does this mean? ). If published, this will include your full peer review and any attached files.

**Do you want your identity to be public for this peer review?** For information about this choice, including consent withdrawal, please see our Privacy Policy .

Reviewer #1: No

Reviewer #2: **Yes: ** Maria Amparo Buaki-Sogo

---

## [Author Response · Author response to Decision Letter 1]

30 Jan 2025

Editors,

PLOS One,

Dear sir/madam,

Author’s response for editorial and reviewer comments

Thank you for your rapid response, encouragements and helpful comments on our submission numbered PONE-D-24-26866 and titled “Comparing membrane sweep and cervical massage in preventing the need for formal labour induction in uncomplicated pregnancy at term: Secondary analysis of a randomized controlled trial”.

We have accepted and addressed all the points raised by the editorial team and the esteemed reviewers.

Amendments made to fulfill journal requirements,

1. Comment –

Response –

Thank you. We have edited the manuscript to comply with the journal style requirements stipulated in the given guidelines.

2. Comment –

We note that the original protocol file you uploaded contains a confidentiality notice indicating that the protocol may not be shared publicly or be published. Please note, however, that the PLOS Editorial Policy requires that the original protocol be published alongside your manuscript in the event of acceptance. Please note that should your paper be accepted, all content including the protocol will be published under the Creative Commons Attribution (CC BY) 4.0 license, which means that it will be freely available online, and any third party is permitted to access, download, copy, distribute, and use these materials in any way, even commercially, with proper attribution.

Therefore, we ask that you please seek permission from the study sponsor or body imposing the restriction on sharing this document to publish this protocol under CC BY 4.0 if your work is accepted. We kindly ask that you upload a formal statement signed by an institutional representative clarifying whether you will be able to comply with this policy. Additionally, please upload a clean copy of the protocol with the confidentiality notice (and any copyrighted institutional logos or signatures) removed.

Response –

Thank you. This study was conducted fulfill the research component of the principal author’s training program leading to specialist qualification and board certification as a consultant obstetrician and gynaecologist. The Board of Study of Obstetrics and Gynaecology, Postgraduate Institute of Medicine (PGIM), University of Colombo Sri Lanka is a higher educational institution and it does not impose restriction on the studies conducted to fulfill the research requirement of the program. The protocol was assessed by two independent examiners appointed by the PGIM. Ethical clearance was granted by the Ethics Review Committee of the Faculty of Medicine, University of Colombo, Sri Lanka. There is no institutional restriction for making the anonymized study data and protocol to be published. The study data and the protocol are solely owned by the authors, and it was a self-funded study. The initial decision made by the authors was to make the anonymised data and the study protocol available upon reasonable request. We are happy to make the same available for public access under CC BY 4.0 in a data repository, and are now available in FigShare which is one of the recommended repositories suggested by PLOS One.

• Madugalle, Sameera; Jayawardane, Indu Asanka; Jayasundara, Chandana (2025). Data set multi.xlsx. figshare. Dataset. https://doi.org/10.6084/m9.figshare.28243112.v1

• Madugalle, Sameera; Jayawardane, Indu Asanka; Jayasundara, Chandana (2025). Data set primi.xlsx. figshare. Dataset. https://doi.org/10.6084/m9.figshare.28243115.v1

• B, Madugalle T. M. S. S.; S, Jayasundara D. M. C.; A, Jayawardane I. (2025). Questionnaire data excel version. figshare. Dataset. https://doi.org/10.6084/m9.figshare.28248005.v1

• S, Jayasundara D. M. C.; B, Madugalle T. M. S. S.; A, Jayawardane I. (2025). Study protocol - Comparing Effectiveness of and maternal Acceptability with uSing cErvicaL massage vs mEmbrane Sweep for cervical ripening in pregnant women at 38th week of geStation at a tertiary care unit. figshare. Online resource. https://doi.org/10.6084/m9.figshare.28310540.v1

We have uploaded a letter signed by the principal author clarifying the author’s decision to make the anonymised study data and study protocol public.

We have also removed the confidentiality notice pertaining to the publication of the study protocol and anonymized data and uploaded a clean copy.

3. Comment

We are unable to open your Supporting Information file [Data set primi.sav and Data set multi.sav]. Please kindly revise as necessary and re-upload.

Response

The .sav files contain the anonymised data of the study. These are save files that can be opened by IBM SPSS statistical software. We also have extracted the data onto excel files and uploaded together with the .sav files. In the response to comment Number 2, we have given the citations for the data sets. The excel files and the original SPSS .sav files have been uploaded as a part of resubmission.

4. Comment

We note that your Data Availability Statement is currently as follows: [All relevant data are within the manuscript and its Supporting Information files.]

Response

We have uploaded the study protocol and anonymised data sets as supporting files and have made them available in Figshare. We have provided links in the response to comment number 2.

We confirm at this time that we have submitted all the raw data required for replication of the study.

5. Comment

Please amend your authorship list in your manuscript file to include authors Dr. Thennakoon Mudiyanselage Salila Sameera Bandara Madugalle.

Response

Thank you. We have included Dr. Thennakoon Mudiyanselage Salila Sameera Bandara Madugalle’s name as surname with initials and he is the Principal and corresponding author.

6. Comment

Your ethics statement should only appear in the Methods section of your manuscript. If your ethics statement is written in any section besides the Methods, please move it to the Methods section and delete it from any other section. Please ensure that your ethics statement is included in your manuscript, as the ethics statement entered into the online submission form will not be published alongside your manuscript.

Response

Thank you. We have moved the ethical statement from the end of the manuscript to the end of the methods section.

7. Comment

Please include captions for your Supporting Information files at the end of your manuscript, and update any in-text citations to match accordingly.

Response

Thank you. We have renamed the supporting information files according to the provided guidance. We have changed in-text citations accordingly.

8. Comment

Response

Thank you. We have rechecked the references list and confirmed that it is complete and correct. There are no retracted papers citied.

Amendments were actioned in response to reviewer comments.

Reviewer 1

1. Comment

In the Method section, please provide descriptions of statistical methods used for comparing outcome rates between groups.

Response

Thank you. We have added descriptions of statistical methods used to compare all the outcome rates between groups in the methods section.

2. Comment

Please correct “long-Rank mantel-Cox test” as “log-rank test”. Please note that it’s log-rank instead of long-rank. Since “log-rank test” is sometimes called “Mantel-Cox test”, you don’t have to put both names together.

Response

Thank you. We have made the corrections.

3. Comment

What’ RR and OR? How were they calculated? Please define acronyms the first time they appear in the text.

Response

Thank you. RR stands for relative risk and OR stands for odds ratio. Methods section contains a brief description of using contingency tables to calculate odds ratio and relative risk. We have defined acronyms at their first use with apologies in the resubmission – Line 194-195

4. Comment

In the results, it is better to use “RR=” and “95% CI=” rather than “RR-” or “95% CI-”.

Response

Thank you. We have made the suggested changes.

5. Comment

“due to small subgroup size” is not the reason for insignificant difference since multiparous group has even smaller size but with significant difference.

Response

Thank you. We have rephrased it.

6. Comment

In table 10, please add p values for each outcome. Those “0” p-values for comparing 0% vs. 0% are not correct. They should be close to 1.

Response

Thank you. We have included the missing probability values with apologies.

Reviewer 2

1. Comment

Abstract is longer than recommended 300 words and would benefit from having headings

Response

Thank you. We have rephrased and edited the abstract to conform with the journal requirements.

2. Comment

Some grammar amendments to improve readability

Response

Thank you. We have tweaked the text to make it more readable and grammatically accurate.

Thank you.

T. M. S. S. B. Madugalle

Principal and corresponding author

---

## [Decision Letter · Decision Letter 1]

2 Mar 2025

PONE-D-24-26866R1Comparing membrane sweep and cervical massage in preventing the need for formal labour induction in uncomplicated pregnancy at term: secondary analysis of a randomized controlled trialPLOS ONE

Dear Dr. Madugalle,

Thank you for submitting your manuscript to PLOS ONE. After careful consideration, we feel that it has merit but does not fully meet PLOS ONE’s publication criteria as it currently stands. Therefore, we invite you to submit a revised version of the manuscript that addresses the points raised during the review process.

We look forward to receiving your revised manuscript.

Kind regards,

Nishel Mohan Shah, PhD

Academic Editor

PLOS ONE

Journal Requirements:

Reviewers' comments:

Reviewer's Responses to Questions

**Comments to the Author**

1. If the authors have adequately addressed your comments raised in a previous round of review and you feel that this manuscript is now acceptable for publication, you may indicate that here to bypass the “Comments to the Author” section, enter your conflict of interest statement in the “Confidential to Editor” section, and submit your "Accept" recommendation.

Reviewer #1: (No Response)

Reviewer #2: All comments have been addressed

2. Is the manuscript technically sound, and do the data support the conclusions?

Reviewer #1: (No Response)

Reviewer #2: Yes

3. Has the statistical analysis been performed appropriately and rigorously? 

Reviewer #1: (No Response)

Reviewer #2: Yes

4. Have the authors made all data underlying the findings in their manuscript fully available?

Reviewer #1: (No Response)

Reviewer #2: Yes

5. Is the manuscript presented in an intelligible fashion and written in standard English?

Reviewer #1: (No Response)

Reviewer #2: Yes

6. Review Comments to the Author

Reviewer #1: “probability values” doesn’t mean “P-values” from a statistical test. You should not say “P value at 95% CI” since the P value is not obtained at 95% CI.

Please ask a biostatistician to help with professional writeup/edits in statistics methods and analysis results. The following are some examples for revisions/corrections.

Lines 180-181, the sentence “…groups were compared using standard deviation, standard error of the mean, and t-tests to calculate probability (p) values at 95% confidence intervals.” Doesn’t described in a professional way. It may be revised as “…groups were compared using t-tests and P-values with 95% confidence intervals are reported.”

The title of Table 7 should be revised as “P-values from Log rank tests for comparisons of “survival” curves after Membrane sweep (MS) versus Cervical massage.” Through the text, since the studied end point is time to spontaneous labor rather than survival, to avoid confusion, I would replace the term “survival” without spontaneous labor using “Proportion of subjects without spontaneous labor”.

In Table 8, “P value at 95% CI” should be corrected as “P value”.

Lines 224-225, “Subgroup analysis according to parity did not show any statistically significant difference between any methods due to the small individual sub-group size.” I have mentioned last time, the reason of insignificance might not be due to small sample size. To be informative, please add more contents to Table 6 with the data in subgroups of parity to show if there are any potential differences in outcomes by parity.

Lines 258-260, the conclusion based on overall data may be misleading. To give complete information, please add sentences to state that the shorter period only be significant in multiparous group.

Reviewer #2: (No Response)

7. PLOS authors have the option to publish the peer review history of their article (what does this mean? ). If published, this will include your full peer review and any attached files.

**Do you want your identity to be public for this peer review?** For information about this choice, including consent withdrawal, please see our Privacy Policy .

Reviewer #1: No

Reviewer #2: No

---

## [Author Response · Author response to Decision Letter 2]

13 Mar 2025

12/03/2025

Editors,

PLOS One,

Dear sir/madam,

Rebuttal letter in response to editorial and reviewer comments - PONE-D-24-26866R1

Thank you for your rapid response, encouragements and helpful comments on our submission numbered PONE-D-24-26866R1 and titled “Comparing membrane sweep and cervical massage in preventing the need for formal labour induction in uncomplicated pregnancy at term: Secondary analysis of a randomized controlled trial”.

We have accepted and addressed all the points raised by the editorial team and the esteemed reviewers.

• Comment - Please ask a biostatistician to help with professional writeup/edits in statistics methods and analysis results.

Response – Thank you. We have obtained the help of a biostatistician to make amendments/editions to the article.

• Comment - Lines 180-181, the sentence “…groups were compared using standard deviation, standard error of the mean, and t-tests to calculate probability (p) values at 95% confidence intervals.” Doesn’t described in a professional way. It may be revised as “…groups were compared using t-tests and P-values with 95% confidence intervals are reported.”

Response – Thank you. Amendments have been made in lines 189-194.

• Comment - The title of Table 7 should be revised as “P-values from Log rank tests for comparisons of “survival” curves after Membrane sweep (MS) versus Cervical massage.”

Response – Thank you. Edited.

• Comment - Through the text, since the studied end point is time to spontaneous labor rather than survival, to avoid confusion, I would replace the term “survival” without spontaneous labor using “Proportion of subjects without spontaneous labor”.

Response – Thank you. Editions made in table 4, Lines 200, 242, Fig 2 and 3 titles

• Comment - In Table 8, “P value at 95% CI” should be corrected as “P value”.

Response – Thank you. Edited.

• Comment - Lines 224-225, “Subgroup analysis according to parity did not show any statistically significant difference between any methods due to the small individual sub-group size.” I have mentioned last time, the reason of insignificance might not be due to small sample size. To be informative, please add more contents to Table 6 with the data in subgroups of parity to show if there are any potential differences in outcomes by parity.

Response – Thank you. Editions have been made. Additional data for Primi and multi subgroups have been included in Table 6.

• Comment - Lines 258-260, the conclusion based on overall data may be misleading. To give complete information, please add sentences to state that the shorter period only be significant in multiparous group.

Response – Thank you. Corrections made in line 321.

Thank you,

T. M. S. S. B. Madugalle

Principal and corresponding author

---

## [Decision Letter · Decision Letter 2]

23 Apr 2025

Comparing membrane sweep and cervical massage in preventing the need for formal labour induction in uncomplicated pregnancy at term: secondary analysis of a randomized controlled trial

PONE-D-24-26866R2

Dear Dr. Madugalle,

We’re pleased to inform you that your manuscript has been judged scientifically suitable for publication and will be formally accepted for publication once it meets all outstanding technical requirements.

Kind regards,

Nishel Mohan Shah, PhD

Academic Editor

PLOS ONE

Additional Editor Comments (optional):

Reviewers' comments:

Reviewer's Responses to Questions

**Comments to the Author**

1. If the authors have adequately addressed your comments raised in a previous round of review and you feel that this manuscript is now acceptable for publication, you may indicate that here to bypass the “Comments to the Author” section, enter your conflict of interest statement in the “Confidential to Editor” section, and submit your "Accept" recommendation.

Reviewer #1: All comments have been addressed

2. Is the manuscript technically sound, and do the data support the conclusions?

Reviewer #1: (No Response)

3. Has the statistical analysis been performed appropriately and rigorously? 

Reviewer #1: (No Response)

4. Have the authors made all data underlying the findings in their manuscript fully available?

Reviewer #1: (No Response)

5. Is the manuscript presented in an intelligible fashion and written in standard English?

Reviewer #1: (No Response)

6. Review Comments to the Author

Reviewer #1: (No Response)

7. PLOS authors have the option to publish the peer review history of their article (what does this mean? ). If published, this will include your full peer review and any attached files.

**Do you want your identity to be public for this peer review?** For information about this choice, including consent withdrawal, please see our Privacy Policy .

Reviewer #1: No

---

## [Editor Report · Acceptance letter]

PONE-D-24-26866R2

PLOS ONE

Dear Dr. Madugalle,

I'm pleased to inform you that your manuscript has been deemed suitable for publication in PLOS ONE. Congratulations! Your manuscript is now being handed over to our production team.

Kind regards,

on behalf of

Dr. Nishel Mohan Shah

Academic Editor

PLOS ONE